# Conflict reducing practices in evolution education are associated with increases in evolution acceptance in a large naturalistic study

Rahmi Qurota Aini[1], K. Supriya[2], Hayley Dunlop[3], Baylee Edwards[4], Samantha Maas[4], Julie Roberts[4], Alexa Summersill[5], Yi Zheng[6], Sara Brownell[4]◉, M. Elizabeth Barnes[1]◉*

1 Department of Biology, Middle Tennessee State University, Murfreesboro, TN, United States of America,
2 Center for Education Innovation and Learning in the Sciences, University of California Los Angeles, Los Angeles, CA, United States of America, 3 College of Medicine, Ohio State University, Columbus, OH, United States of America, 4 School of Life Sciences, Arizona State University, Tempe, AZ, United States of America,
5 Psychology Department, Middle Tennessee State University, Murfreesboro, TN, United States of America,
6 School of Mathematical and Statistical Sciences, Arizona State University, Tempe, AZ, United States of America

◉ These authors contributed equally to this work.
* Liz.Barnes@mtsu.edu

**Data Availability Statement:** The dataset and R codes supporting the conclusions of this article is available at GitHub (https://github.com/

## Abstract

Evolution is an important part of biology education, but many college biology students do not accept important components of evolution, like the evolution of humans. Practices that reduce perceived conflict between religion and evolution have been proposed to increase student evolution acceptance. This study investigates college student experiences of conflict reducing practices in evolution education and how these experiences are related to their gains in acceptance of human evolution during evolution instruction. We measured the natural variation in student experiences of conflict reducing practices among 6,719 college biology students in 55 courses and 14 states including (1) their experiences of an instructor demonstrating religion-evolution *compatibility* by presenting examples of religious leaders and scientists who accept evolution and (2) their experiences of an instructor emphasizing students' *autonomy* in their own decision to accept evolution or not. We also measured student acceptance of human evolution before and after instruction so that we could test whether any changes in evolution acceptance were associated with student experiences of the conflict reducing practices. Linear mixed models showed that highly religious Christian students accepted evolution more when they perceived more compatibility practices. Further, students from all religious and non-religious affiliations accepted human evolution more after instruction when they perceived more autonomy practices. These results indicate that integrating examples of religion compatibility in evolution education will positively impact Christian students' views on evolution and that emphasizing students' autonomy over their decision to accept evolution may be important for students more broadly. If instructors incorporate practices that emphasize compatibility and one's personal choice to accept or not accept evolution, then these results suggest that students will leave their college biology classes accepting evolution more. Perhaps by using more conflict reducing practices,

BioedSPSlab/conflict-reducing-practices-quasi-experiment).

**Funding:** This study is funded by National Science Foundation #1818659. The funders had role in data collection.

**Competing interests:** The authors have declared that no competing interests exist.

instructors can help increase evolution acceptance levels that have remained low in the United States for decades.

## Introduction

### The perceived conflict between religion and evolution impacts college student evolution acceptance

Evolution is the foundation of biology [1–3], yet it has remained controversial in the public and in schools since Darwin first published *The Origin of Species* [4]. In the United States, about half of the public rejects human evolution [5, 6] and this rejection also extends to American college biology students. Surveys indicate that in 2020, up to 35% of introductory college biology students in the United States did not think life shares common ancestry [7]. If students are to apply evolution to their biological thinking outside of the classroom, they must not only understand but also accept evolution [8]. Why is a part of biology that is foundational to the discipline so controversial, even among college biology students?

Perceived conflict between evolution and a student's religious beliefs and religious culture has been shown to be one of the biggest factors influencing evolution acceptance [9, 10]. There are substantial and consistent negative correlations between a student's level of commitment to their religion, which is called religiosity, and the extent to which they accept evolution [11, 12]. Low acceptance of evolution is also related to students' specific religious affiliations. In the United States, Christian, and Muslim biology students tend to have the lowest evolution acceptance while Hindu, Jewish, agnostic, and atheist students tend to have the highest evolution acceptance levels [9, 13]. However, there are many students who are highly religious Christians and Muslims who do accept evolution because they do not see conflict between evolution and their religion [10, 14–16]. For instance, some Christian biology students have reconciled their religion and evolution by choosing to interpret the Bible as allegory rather than literal and by seeing examples of Christian biology professors who accept evolution [15]. Although religiosity or a specific religious affiliation is a predictor of evolution acceptance, this relationship is ultimately rooted in the extent to which one perceives conflict with evolution and their specific religious beliefs, cultures, and teachings [9]. Thus, reducing perceived conflict between religion and evolution has been recommended as a potentially effective way to mitigate low acceptance rates of evolution [16–21].

### A potential solution: Cultural competence and conflict reducing practices in evolution education

Despite the potential importance of reducing perceived conflict between religion and evolution, evolution instructors can struggle with helping students reduce their perceived conflict. While the majority of undergraduate students in biology courses are religiously affiliated [22], only about 25% of biologists are religiously affiliated [23, 24], creating a situation where predominantly non-religious instructors are teaching evolution to mostly religious students [25]. Instructors are in positions of power relative to their students and as such, can amplify or mitigate cultural divides between themselves and students [26]. However, research suggests that instructors often teach evolution in ways that are not culturally inclusive for religious students [27, 28], and this may lead to fewer students accepting evolution [27]. For instance, professors teaching evolution at secular institutions often reported being unwilling to implement

instruction that could help reduce students' perceived conflict [28]. Conversely, evolution instructors teaching at Christian institutions reported that they used a variety of strategies to help Christian students become more comfortable with evolution due to their own experiences reconciling their religion and evolution as biologists [14]. Thus, we have advocated that instructors should try to mitigate potential consequences of religious identity differences between themselves and students by practicing cultural competence and using conflict reducing teaching strategies when teaching evolution [25]. Cultural competence is the ability of individuals from one culture (in this case, secular instructors who are teaching evolution) to bridge cultural differences to effectively communicate to individuals from a different culture (in this case, religious undergraduate biology students) [26, 29–31].

Cultural competence has become a useful framework for describing how instructors can aim to bridge divides based on demographic and cultural differences between themselves and their students. This framework was originally born from healthcare studies that emphasized the negative impact of racial/ethnic and socioeconomic difference between mostly white, high income doctors and their racial/ethnically and income diverse patients [32]. This original work emphasized the need of these physicians in positions of authority and power to cultivate an understanding of how to effectively communicate with those who are culturally different from them. Since then, cultural competence was adopted in K-12 education [33] and more recently in undergraduate STEM education to also emphasize bridging cultural divides, but this time between *instructors* and *students* [26]. Our research groups were the first to propose cultural competence as way to bridge religious cultural divides in college biology between mostly secular instructors and their majority religious students while teaching evolution [25]. Since then, we have more specifically referred to practices in evolution education that can be considered religiously culturally competent as conflict reducing practices, to more clearly emphasize that reduction of perceived conflict between religion and evolution is the most relevant religiously culturally competence practice in evolution education at the college level.

Despite broad recommendations for experts to teach evolution using culturally competent and conflict reducing practices [16, 19, 25, 34, 35], there are a lack of studies that collect data on student outcomes before and after instruction as well as studies that compare outcomes of students that experience conflict reducing practices to students that do not (see **S1 Appendix** for a list of studies and their methods). Further, although many researchers recommend conflict reducing practices, some have suggested that avoiding the potential controversy and only teaching to an understanding of evolution is most beneficial for student outcomes [36, 37]. We need studies comparing student outcomes with and without conflict reducing practices to determine which recommendations are most supported by the evidence.

## The current study

In this current study, we compared student evolution acceptance when they perceive different levels of conflict reducing practices in their evolution instruction. We used the natural variation in student experiences of instructor practices to determine whether experiencing conflict reducing practices leads to improved acceptance of evolution among students. We were able to explore student experiences of two practices that could be considered conflict reducing during evolution instruction: 1) showing examples of potential ***compatibility*** between religion and evolution by highlighting religious scientists, leaders, and/or church members who accept evolution [15, 18, 20], and 2) helping students feel a sense of ***autonomy*** over their decision to accept or reject evolution [38, 39] so that they do not feel as if they are being forced into accepting evolution. These practices are hypothesized to be effective because they reduce students' perceptions that their religious identity is a barrier to their acceptance of evolution.

Showing religious individuals who accept evolution may give religious students examples of role models who have reconciled their religion and evolution [15] and may combat negative stereotypes of religious individuals in science that may make students feel like they do not belong in the classroom environment [40, 41]. Giving students a sense of autonomy may reduce perceptions that they are being forced to accept evolution [21] and increase their motivations to be accuracy oriented in their reasoning [38]. Our research questions were:

1. Are students' perceptions of conflict reducing practices during their evolution instruction related to their evolution acceptance after instruction?

2. Is the effect of these conflict reducing practices different for Christian students, other religious students, and non-religious students?

## Materials and methods

Between fall 2018 and spring 2021, we surveyed a population of undergraduate biology students from introductory biology courses in 14 states (CA, AZ, UT, TX, OK, MN, WI, MI, NY, NC, SC, AL, FL, and HI) before and after they learned evolution. Before learning evolution, we measured students' human evolution acceptance, evolution understanding, religious affiliation, and religiosity, as well as other demographic control variables. After evolution instruction, we again measured student acceptance of human evolution. We also measured students' level of agreement that they experienced compatibility through examples of religious individuals who accept evolution and that the instructor gave the students sense of autonomy in their decision to accept evolution. **Fig 1** depicts our study design and details of each measure are described below.

### Ethical considerations

Arizona State University's Institutional Review Board approved this study (protocol no. 8191). Students and instructors in this study filled in the consent form before participating in the study.

### Measures and validity evidence

**The outcome measure: Evolution acceptance.** We used the Inventory of Student Evolution Acceptance (I-SEA) to measure students' human evolution acceptance [42]. This measure

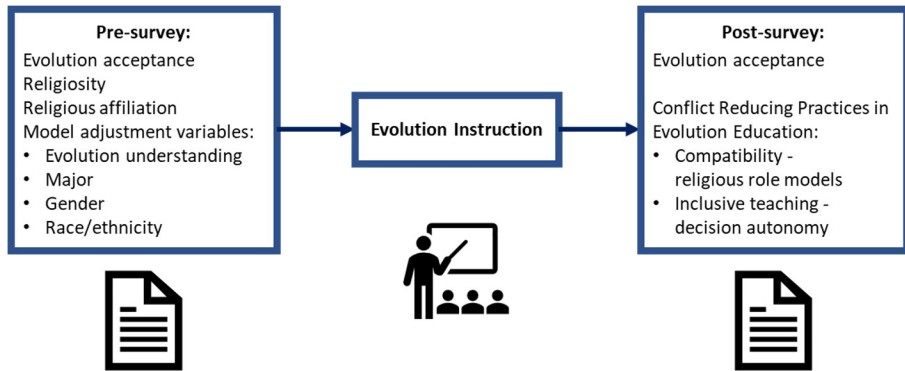

**Fig 1. Illustration of study design for exploring the impact of conflict reducing evolution education practices on student acceptance of evolution.**

includes statements with which students agree or disagree on a five-point scale. There are 8 items that measure acceptance of human evolution (e.g., "like other organisms, the human species is a result of evolution from an ancestral group"), 8 items that measure acceptance of macroevolution (e.g., "I think that new species arise from ancestral species"), and 8 items that measure acceptance of microevolution (e.g., "natural selection is a reasonable explanation that describes the ways in which groups of organisms have changed over time"). Among college biology students, confirmatory factor analyses and Rasch analyses of I-SEA data confirm items on the I-SEA fall onto these three separate psychological constructs [43, 44]. Students typically have lower levels of acceptance of human evolution and macroevolution compared to microevolution due to higher levels of perceived conflict with religion for human evolution and macroevolution [42, 45–48]. We chose to focus on human evolution acceptance when reporting results of this study since human evolution tends to elicit the most perceived conflict with religion.

**Predictor variables: Conflict reducing practices in evolution education.** *Survey item development*. We developed survey items to measure the extent to which students experienced conflict reducing practices in their evolution instruction. We surveyed *students* about their experiences directly to use as predictors of their outcomes, rather than relying on *instructor* or *researcher* reports of classroom practices, because we wanted to gauge the efficacy of successful implementation of conflict reducing practices that students actually experienced. If instructors perceived that they did something that students did not notice, we predicted that this would not affect student experiences, so we relied on student perceptions. This approach is best suited to the naturalistic study design such as this in which students were learning evolution from many different instructors who were implementing widely different evolution education practices. The ways by which instructors decided to present examples of compatibility and give students autonomy may be differentially experienced by students, both between classes of different instructors but even within the same class with the same instructor. For instance, an instructor could report and/or a researcher could observe that the instructor gave examples of compatibility because in passing the instructor mentioned the religious background of an evolutionary biologist. However, students may not have heard the instructor due to absence, disengagement, or even momentary distraction. Thus, even though a researcher or instructor would report that they implemented the instruction, the student may never have been exposed to the instruction. Even students who pay attention to the instruction may experience the instruction in different ways. For example, an instructor may present religious scientist role models by showing a Hindu scientist who accepts evolution; this may be experienced as examples of compatibility by Hindu students but may not be perceived as compatibility for *Christian* students. Further, more religious students may need to hear more substantial amounts of explicit instructor talk to perceive they are being given autonomy over their decision to accept evolution. There are many different ways that instructors can implement conflict reducing practices and students can also perceive practices in many different ways. In this study, we were most interested in relating the extent to which students *experience* conflict reducing practices with their evolution education outcomes. Thus, we measured students' experiences of conflict reducing practices by using students' agreement and associating that agreement with their evolution acceptance after instruction.

We developed items based on prior literature illustrating conflict reducing practices [15, 17, 19–21, 27, 35, 39, 49], which resulted in 14 items. Based on this prior literature, these items broadly covered instructional practices that illustrated compatibility between religion and evolution, such as describing the bounded nature of science and presenting examples of religious scientists and leaders who accept evolution. It also included items that we considered measured inclusive teaching practices for religious students such as encouraging student

exploration of their views on religion and evolution and remaining respectful of religious beliefs. All of the original 14 items can be found in **S1 Appendix**. Students responded to each item with their agreement on a 6 pt. Likert scale (strongly disagree–strongly agree) (see **S1 Text**).

*Expert and student review*. The initial items were revised based on the feedback of a panel of five biology education researchers who reviewed the items for content validity and readability. To establish response process validity evidence [50], we conducted cognitive interviews using think-aloud methods [51] with 25 undergraduate biology students and items were revised for readability based on the interview responses.

*Construct validation*. To provide internal validity evidence for the measurement [50] of conflict reducing practices, we used a combination of exploratory and confirmatory factor analysis. We reviewed all 14 items and based on theoretical distinctions between items we hypothesized a two-factor model of items representing (1) compatibility instruction and (2) inclusive teaching of religious students. However, a confirmatory factor analysis of data from our first data collection indicated that this model was a poor fit (CFI < 0.95, RMSEA > 0.06) so we used an exploratory factor analysis to explore the structure of the data further and reviewed the items again to identify if we could build a more robust model. The scree plot and factor loadings indicated there were two factors, but some items had lower factor loadings than other items (<0.70) and upon inspection of these items we identified theoretical differences. For instance, the 4 items within "compatibility" that had the strongest loadings were all items regarding the extent to which the instructor showed examples of religious people and scientists who accepted evolution (for instance, the item "I felt like the instructor helped me realize that there are scientists who accept evolution and are also religious" had a factor loading of 0.97) while other items within this factor that had lower loadings were slightly different (for instance, the item "I felt like the instructor indicated that evolution and religion can be compatible" had a factor loading of 0.64). Similarly, only three items within the "inclusive teaching" had strong factor loadings and these items all referred to students' decision autonomy (for instance, "I felt like the instructor let students make up their own mind about evolution."). We identified 7 items that were a poor fit and decided to remove them and then re-operationalize our constructs based on the remaining items. We revised "compatibility" to more specifically refer to "compatibility through religious role models" and was defined as "the extent to which students agreed that the instructor helped them realize there are religious leaders, scientists, and church members who also accept evolution." We revised "inclusive teaching" to more specifically refer to "inclusive teaching through decision autonomy" and was defined as "the extent to which students agree that the instructor gave them a sense of control over their decision making about evolution acceptance". We ran a subsequent EFA and found that there were again two factors of items with each loading >0.70 on this round of analysis. We then used these remaining 7 items to collect data in subsequent semesters and ran another CFA on the aggregated data resulting in a model with an acceptable fit (CFI = 0.981, RMSEA = 0.040). All final items used in the analyses can be found in in **S1 Text**.

*Class observations*. Although we were primarily interested in student subjective experiences of the instruction, we still wanted to confirm that student agreement with survey items corresponded to instructional practices in the classroom to establish criterion validity evidence for the survey [50]. So, we compared student reports on survey items to observations from the classroom instruction. Six participating instructors agreed to share recordings of their evolution instruction and provide the research team with course artifacts such as syllabi, homework assignments, and PowerPoints from their evolution instruction. M.E.B. analyzed students' data from each class and categorized classes in which more than half of students agreed that a practice was occurring as "practice present" and classes in which more than half of students disagreed that a practice was present as "practice absent". Independently, H.D., J.R., and S.M.

reviewed all course artifacts and coded each course as "practice present" or "practice absent". The three research assistants wrote summaries of their observations, met and discussed any discrepancies between their coding, and came to agreement on final codes. We then compared codes based on student reports to codes based on researcher classroom observations. We found 92.3% agreement between student reports and researcher observations. This gave us confidence that despite the variation we will see in student experiences within each course that the majority of students (more than 50%) would agree with general researcher observations of the instruction.

*Limitations of final measures.* Our final measure of compatibility instruction only included items on student perceptions of instructor use of examples of religious scientists, leaders, and church members who accept evolution to illustrate the potential compatibility between religion and evolution. However, illustrating potential compatibility between religion and evolution has been described in other ways including discussing the bounded nature of science [52–55] and discussing the spectrum of viewpoints on compatibility between evolution and religion [17, 49, 56, 57]. Further, our final measure of inclusive teaching of religious students only included three items referring to students' perceptions that the instructor "let them make up their own mind about evolution" or "wasn't trying to force students to accept evolution". However, our original conception of this practice included encouraging exploration of personal views [49, 58–60] and remaining respectful towards students who do not accept evolution. Thus, this study will not reveal the potential impacts of these other instructional practices that could illustrate compatibility and inclusive teaching. From this study, we learned that measuring these specific practices using a student survey is limited because student perceptions of these practices may be correlated with one another but fall on ambiguous dimensions when trying to establish the structural validity of the measures.

*Rasch transformation.* We used Rasch modeling to convert ordinal Likert scale scores into equivalent interval logit scale measures to account for varying psychological distances between Likert scale options. The procedure generated a person ability score for each participant for each measure (known as *person measure*, a higher value indicates a student agreed more compared to other students). In the analyses, we used the person measure scores instead of the average of Likert scores [61, 62]. Discussion of item fit statistics (MNSQ), item-person relationships using Wright Maps and Item Characteristic Curve plots can be found in the **S1 and S2 Figs,** and reliability values in **S2 Table.**

**Predictor variables: Religiosity and religious affiliation of student.** To measure religiosity, we used an instrument originating from the psychology of religion [63]. The instrument consisted of four items with which the students agree or disagree on a five-point scale. The items measure the intrinsic strength of one's religious identity (e.g., "I consider myself a religious person") and participation in religious activities (e.g., "I attend religious services regularly"). These items are similar to other common measures used both in studies of religion [64, 65] and studies of evolution acceptance [11, 66]. To determine students' religious affiliation, we asked students to self-identify from the following list of religious affiliations: agnostic, atheist, Buddhist, Christian–Catholic, Christian–The Church of Jesus Christ of Latter-day Saints, Christian–Protestant, Christian–Other, Christian–nondenominational, Hindu, Jewish, Muslim, nothing in particular, other faith, and decline to state.

**Control variables.** We collected information to use as control variables in our models. Particularly for a naturalistic study design, it is important that we attempt to control potentially confounding variables in our statistical analyses. First, we included students' course as a nested variable in our model to control for the possibility that non-independence of data points within each class could influence results of the study. Next, we controlled for students' level of acceptance of evolution at the start of evolution instruction because students who come in

with a high acceptance of evolution (1) may be more receptive to evolution instruction, including conflict reducing practices and (2) may have less acceptance to gain than students who start instruction with lower levels of evolution acceptance. We also collected students' level of understanding of evolution [67] before instruction to control for the possibility that students who perceived more conflict reducing practices also had a higher understanding of evolution than students who did not perceive them. We also controlled for whether the student was a biology major, their gender, and race/ethnicity because these variables are often related to acceptance of evolution [66, 68–70] and we wanted to reduce the possibility that uneven distribution of these variables across courses with different levels of conflict reducing practices may explain any statistically significant relationships between the instruction and student outcomes.

## Sample

**Sampling method.** Our targeted sample was students in undergraduate introductory biology classes. We purposefully targeted a broad sample and used convenience and snowball sampling methods. We recruited instructors initially from the listserv for the Society for the Advancement of Biology Education Research (SABER), which is mostly comprised of faculty and graduate students who teach college level biology. The initial recruitment explained that we were interested in determining what college level evolution education practices were effective for improving student outcomes and that we would use the natural variation in instructor practices to determine effective practices. Since we wanted to recruit instructors who were both using and not using conflict reducing practices, we did not mention our specific interest in exploring conflict reducing practices or student evolution acceptance in the recruitment email. Instead, we only indicated we were interested in identifying evidence-based practices for teaching evolution effectively. Some instructors we recruited then recommended to us their colleagues who were teaching evolution in introductory biology. In subsequent semesters, to increase the number of courses that could be included in our analyses, we identified large enrollment introductory biology classes and emailed the instructors to ask for their participation. We recruited primarily from public doctoral granting universities. Students were offered a small amount of extra credit for completing the surveys.

**Participants.** A total of 6,719 students from 55 courses completed both the pre and post instructional surveys and were included in the final analyses (~ 44% response rate). A 44% response rate is within the typical average for online surveys [71] and is particularly acceptable for a longitudinal study with data collected at multiple time points. The nature of the response rate for the survey is also influenced by various factors such as students' interest in the topic, survey structure, communication methods, or incentives for participation [72]. We also implemented an honesty check at the end of the survey to assess whether students were paying attention and providing accurate responses. This step helped ensure the reliability of the data collected. In the analyses, students are nested by course, so we removed any courses from the data set in which there were less than 20 complete student responses as recommended by [73]. Class size and response rates from each class can be found in **S1 Table**.

We categorized student religious affiliation into three groups based on sample size considerations: Christian, other religion (e.g., Hindu, Buddhist, Jewish, Muslim), and no-religion (e.g., Atheist and Agnostic). Then we categorized Christian students and students from other religions based on their religiosity levels. Students who scored 1 standard deviation above the average on the religiosity scale were categorized as "high religiosity" and students who scored between -1 to 1 standard deviations on the religiosity scale were categorized as "average religiosity". Students who scored more than 1 standard deviation below the average on the religiosity scale were recategorized to belong to the "no-religion" group. All of these students were

confirmed to have disagreed on the religiosity survey that they were religious or that religion was important to their lives. So, although these students may have checked the box for "Christian", "Muslim", "Jewish", etc, we considered them non-religious because they did not believe this was a relevant or important part of their identities. A breakdown of the sample demographics can be found in Table 1.

## Sampling limitations

Even though we were ambiguous in our email recruitment about the specific focus of our study, instructors still self-selected into this study and may not represent the broader population of evolution instructors. However, our research questions of interest were about the

**Table 1. Sample of students used in analyses broken down by gender, race/ethnicity, major, and semester of data collection.**

| Total | (n = 6,719) |
|---|---|
| *Gender* | |
| Man | 29.2% |
| Woman | 70.8% |
| Non-binary[a] | – |
| *Race/ethnicity* | |
| PEER[1] | 21.2% |
| Asian | 18.3% |
| white | 51.0% |
| multiracial | 9.5% |
| *Major* | |
| Biology Major | 54.8% |
| Non-biology Major | 45.2% |
| *Semester* | |
| fall 2018 | 19.7% |
| fall 2020 | 25.0% |
| spring 2020 | 30.0% |
| spring 2021 | 25.3% |
| *Religion* | |
| No-Religion[2] | 31.7% |
| Christian[3] | |
| high religiosity | 16.3% |
| average religiosity | 38.8% |
| Other Religion[4] | |
| high religiosity | 1.2% |
| average religiosity | 13.0% |

[1] We chose to collapse these groups due to the limitations of the sample sizes of each individual group. PEER (persons excluded based on ethnicity or race) students included: American Indian, Native American, or Alaskan Native, Black or African American, Hispanic or Latinx, and Native Hawaiian or Other Pacific Islander.

[2] atheist, agnostic, and students with low religiosity from Christian, Muslim, and other groups

[3] Christian–Catholic, Christian–The Church of Jesus Christ of Latter-day Saints, Christian–Protestant, Christian–Other, Christian–nondenominational

[4] We chose to collapse these groups due to the limitations of the small sample sizes of each individual group. Other religion students include Buddhist, Hindu, Jewish, and Muslim.

[a] We had 44 non-binary students in our data set, but we did not include them in the analysis due to low sample size and the lack of prior data on how gender including non-binary students may affect evolution acceptance.

relationships between variables and not the generalizability of the descriptive statistics of the sample. Nonetheless, it is possible that instructors who were interested in participating in an evolution education study may be more likely to implement conflict reducing practices such as discussing examples of potential compatibility between religion and evolution. Further, it is possible that students in these classes we surveyed or those who chose to respond to the survey may not be representative of biology students more broadly. Although we cannot explore all ways the sample may be unique from the general population, we did compare key demographic characteristics of our sample with data from the National Science Foundation's National Center for Science and Engineering Statistics [74] to assess how closely our sample aligns with the broader student population. Note that our sample consisted of students enrolled in introductory biology courses and only 54% of whom were biology majors, and therefore we compared with national data on degrees earned in science and engineering fields. The racial and ethnic composition of our sample is similar to national data. For instance, 51% of our sample identified as White (compared to 58% nationally), 18% identified as Asian (compared to 12% nationally), and 21% identified as PEER (Persons Excluded because of Ethnicity or Race), compared to 26% in the broader population. For gender composition, 70% of our sample identified as women, which is consistent with national trends. In 2020 [74], women earned 66% of bachelor's degrees in science and engineering with a particularly high representation in fields like psychology (where women accounted for 79% of degrees earned). While we cannot claim full representativeness, these comparisons suggest that our sample broadly mirrors the diversity of the student population in relevant ways. It is also important to acknowledge the sample size limitations for students from non-Christian religious groups (e.g., Buddhist, Hindu, Jewish, Muslim), which led us to collapse these groups into a broader "Other-Religion Students" category. While this approach allowed us to maintain sufficient statistical power, it limits our ability to address the distinct experiences and perspectives of each individual religious group. Consequently, our discussion may not fully capture the unique nuances specific to students from smaller religious affiliations.

## Analyses

To analyze the associations of instruction with student outcomes and whether the associations of instruction with outcomes depended on students' religiosity and religion, we used multilevel models with course as a random effect [75].

**Main effect of instruction on evolution acceptance.** This primary analysis examines the direct effect of instruction (compatibility and autonomy) on student acceptance of evolution with course as a random effect. As mentioned in the control variables section, we controlled for students' level of understanding and acceptance of evolution before learning evolution, whether the student was a biology major (nonmajor vs major (reference group)), whether the student was a woman (woman vs man (reference group)), and student race/ethnicity (Persons Excluded because of their Ethnicity or Race (PEER) [76], Asian, multiracial, white (reference group). The rationale for these control variables are (1) to account for baseline demographic factors that could influence outcomes and (2) to ensure that results were not due to students who have a higher understanding of evolution perceiving conflict reducing practices more. The following model was analyzed using R software:

*Post-human evolution acceptance ~ Pre-human evolution acceptance + Pre-evolution understanding + Major + Race + Gender + Instruction + (1|course)*

## Interaction effect: Religion, religiosity, and instruction

To explore whether the instructional effects differed across students with different religious affiliations or religiosity levels, we also tested the same models with interactions

between instruction and religiosity, and instruction and religion with course as a random effect:

*Post- human evolution acceptance ~ Pre-evolution acceptance + Pre-evolution understanding + major + Race + Gender + Religiosity\* Instruction + (1|course)*

*Post- human evolution acceptance ~ Pre- human evolution acceptance + Pre-evolution understanding + major + Race + Gender + Religion\* Instruction + (1|course)*

We also explored whether the instruction effects differed across different religious affiliations and religiosity levels using a 3-way interaction model with course as a random effect:

*Post- human evolution acceptance ~ Pre- human evolution acceptance + Pre-evolution understanding + major + Race + Gender + Religion\*Instruction\* Religiosity + (1|course)*

*Probing the interaction effect*. In the manuscript we report coefficients of the instruction variables and their interactions as well as their statistical significance. We also provide figures (Figs 3 and 5) that depict the direction and strength of the relationships between student outcomes and the instruction, broken down by student religion and religiosity levels (Average religiosity group, *-1SD to +1 SD;* High religiosity group $\geq$ +1 SD). We used the functions *interact_plot* to generate figures and *sim_slopes* to probe interaction effects using Johnson-Neyman intervals from the interactions package in R [77]. These analyses determine if the relationships between independent variables (religion and religiosity) and the focal predictor (instruction) is statistically significant [78]. Full regression tables with coefficients, standard errors, and exact p-values for all analyses reported in this manuscript are listed in **S1 Appendix.**

## Results

### Finding 1: After highly religious Christian students experienced compatibility instruction, they accepted evolution more than highly religious Christian students that did not experience that instruction

**Student perceptions of instruction.**   Overall, we found that students in this sample perceived moderate levels of compatibility instruction by including examples of religious individuals who accept evolution (M = 3.77, SD = 0.69, Range = 1–6) but there was variation in student experiences both within and between classes (Fig 2).

**Main effects.**   Linear mixed models predicting students' evolution acceptance after instruction showed that students' human evolution acceptance increased the more students agreed that the instructor showed examples of religious scientists, leaders, and others who accept evolution (b = 0.01, SE = .00, *p* < 0.001). *Interactions and simple slopes*. We found a significant two-way interaction between students' religiosity and instruction (b = 0.01, SE = 0.00 *p* < 0.001) indicating that this compatibility instruction was more effective for students who were more religious. We also found a significant two-way interaction between religion and instruction for Christian students (b = 0.04, SE = 0.01, *p* < 0.001), indicating that this instruction was particularly effective for Christian students. Considering both religion and religiosity with instruction, we found a significant three-way interaction for Christian students only (b = 0.01, SE = 0.01, p < .05). Simple slope analyses are reported in **Fig 3** and revealed that compatibility instruction was only significantly related to the outcomes of highly religious Christian students in which one logit increase in their agreement that religious role models were present in instruction was associated with a .04 point increase in Christian students' evolution acceptance after instruction.

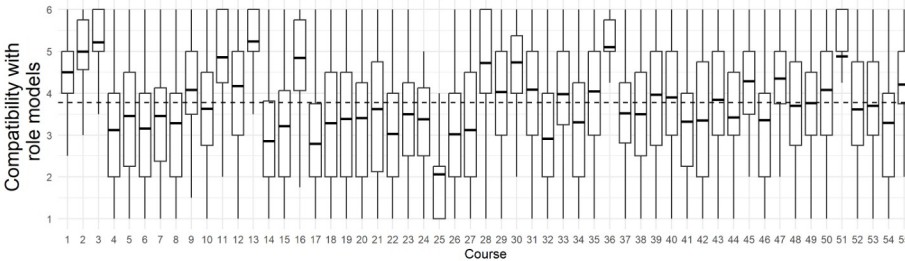

**Fig 2. Student agreement that instructors showed compatibility between religion and evolution by providing examples of religious scientists, leaders, and others who accept evolution, disaggregated by course.** Each box in the plot represents the interquartile range (IQR) of the compatibility scores, with the vertical whiskers representing the spread of maximum and minimum scores, and line inside the box indicating the average score for each course. The dashed horizontal lines represent the overall average score within all samples. The plot is designed to provide a clear view of the distribution and central tendency of the compatibility scores within and between different courses.

## Finding 2: After students experienced instruction that was inclusive and gave them autonomy over their decision to accept evolution, they accepted evolution more than students who did not experience that instruction

**Student perceptions of instruction.** Students in this sample perceived relatively high levels of autonomy during their evolution instruction. Across all courses students tended to agree that instructors gave them autonomy over their decision to accept evolution (M = 4.79, SD = 0.26, Range = 1–6). Fig 4 illustrates the variation in student reported experiences within and between different classes.

**Main effects.** Variation in students' perception of autonomy in their instruction was significantly and positively related to their evolution acceptance after instruction (b = 0.08, SE = 0.01, $p < 0.001$) indicating that for each logit increase in student agreement that autonomy instruction was present, there was a .08 point increase in student evolution acceptance after instruction. *Interactions and simple slopes.* Two-way interactions between religion and instruction and religiosity and instruction were not significant for students' human evolution acceptance ($p > 0.24$), and the three-way interactions between religion, religiosity, and

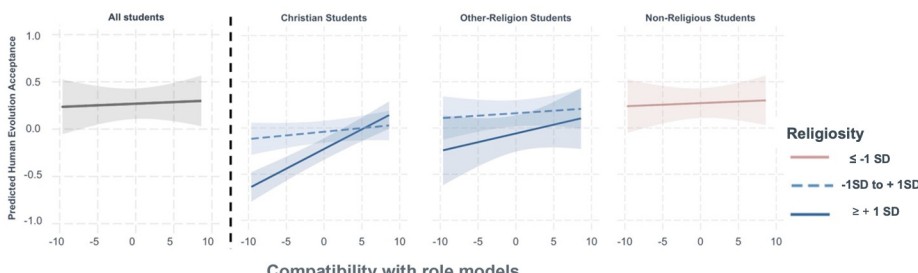

**Fig 3. Illustration of interaction effects from linear mixed models of student religion and religiosity levels on the association between religion negativity during instruction and students' human evolution acceptance after instruction.** The red lines are students who identify with no religious affiliation and students with religiosity measures more than 1 SD below the mean, which we have coded as having no religious affiliation. Values are based on model prediction of 3-way interaction between instruction, religion, and religiosity for human evolution acceptance. The shaded area represents the 95% confidence interval (CI) for the linear model. The simple slope analyses were significant only for highly religious Christian students (β = 0.04, SE = 0.01, $p < 0.001$).

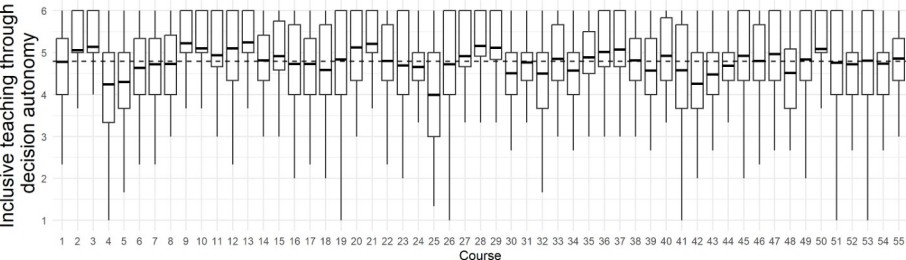

**Fig 4. Student agreement that instructors were implementing inclusive teaching about evolution by giving students autonomy over their decisions to accept evolution, disaggregated by the 55 courses.** Each box in the plot represents the interquartile range (IQR) of the autonomy scores, with the vertical whiskers representing the spread of maximum and minimum scores, and line inside the box indicating the average score for each course. The dashed horizontal lines represent the overall average score within all samples. The plot is designed to provide a clear view of the distribution and central tendency of the autonomy scores across different courses.

instruction for students' evolution acceptance were also not statistically significant ($p > 0.06$) indicating that inclusive teaching with decision autonomy was equally effective for students from all religiosity and religion groups (Fig 5).

## Discussion

In this large study using students sampled across the nation, we found that when students perceive conflict reducing practices during their evolution instruction, they have better evolution acceptance outcomes at the end of instruction. Below, we discuss these results in more detail and connect our current findings to the broader literature on the effectiveness of conflict reducing practices.

### Student perceptions of conflict reducing practices in undergraduate biology education

Because this was a self-selected sample of instructors and students in which we measured student perceptions of instruction and not actual instruction, we have a limited ability to make claims about the relative abundance of conflict reducing practices in biology education

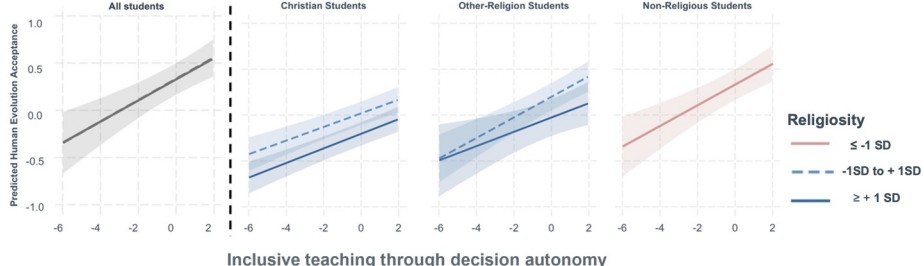

**Fig 5. Illustration of non-significant interaction effects from linear mixed models of student religion and religiosity levels on the association between decision autonomy during instruction and students' human evolution acceptance after instruction.** The red lines are students who identify with no religious affiliation and students with religiosity measures more than 1 SD below the mean, which we have coded as having no religious affiliation. Values are based on model prediction of 3-way interaction between instruction, religion, and religiosity for human evolution acceptance but this was not statistically significant, indicating no statistical differences between groups. The shaded area represents the 95% confidence interval (CI) for the linear model.

broadly. However, based on this data set of 6,719 students in 55 courses and 14 states, there are some important trends in our data worth considering. First, some researchers may have expected that instructors who cared enough to participate in a survey study on how to identify evidence-based practices in evolution education might be more likely to use conflict reducing practices, and students did tend to agree that instructors were giving them autonomy over their decision to accept evolution (Fig 4). But among this sample, students tended to disagree that instructors were giving examples of potential compatibility between religion and evolution (Fig 2). So, this could mean that this conflict reducing practice is not common among instructors. Further, there seemed to be differences between the extent to which students within each class perceived the instruction. In some courses, most students agreed that instructors showed examples of religious leaders and biologists who accept evolution while in other courses most students disagreed that instructors were providing these examples of compatibility, and this indicates that students were perceiving the instruction similarly in these classes. However, there were some courses in which students seemed to have quite different experiences of the instruction from one another. In some courses, a large proportion of students agreed that an instructor helped them realize that religious leaders and religious biologists can accept evolution *and* a large proportion disagreed (Fig 2). It could be that in these instances, the way that the instructor showed examples of compatibility were more subtle and thus were only perceived by some students while others did not notice. It also could be that attendance in these classes was low and these students who disagreed were not exposed to the instruction. Student perceptions of autonomy practices on the other hand, seemed to be more consistent both within and across courses and were consistently high (Fig 4). This could mean that perceptions of autonomy are high in undergraduate evolution education courses or that this sample of instructors were particularly likely to be implementing practices that lead to this perception.

### Are students' perceptions of conflict reducing practices during their evolution instruction related to their evolution acceptance after instruction?

In this study, when highly religious Christian students reported that instructors showed them examples of religious individuals who accept evolution, this was associated with increases in student evolution acceptance. This is in line with prior research that has documented the positive effect of religious role models who accept evolution on religious student evolution acceptance [15, 18, 20, 49]. However, in this study, we did not see a positive effect of these examples for religious students who were not Christian or Christian students who scored low to average on religiosity. This could be due to the religious affiliations of examples presented by instructors and the salience of the identity to students. Social identity theory would predict that students will respond more to examples of religious individuals that have affiliations closer to their own identities and when they themselves strongly associate with the identity [79]. Thus, if instructors were presenting examples of religious leaders and biologists that were apparently Christian, this may have been more impactful for highly religious Christians compared to Christian students who did not report being as religious or compared to non-Christian students. Since we did not have observations of all instruction these students received, we cannot report the affiliations of examples instructors used, but of the courses we did observe that presented examples of compatibility, they were from Christian affiliations. Related, highly religious Christian students may be more likely than other religious students to even notice the representation of Christian role models who accept evolution. Future research could explore the extent to which role models need to reflect closely students' own religious identity and background to be effective for increasing evolution acceptance.

In this study we also have now characterized the practice of decision autonomy and its impact on evolution acceptance, which we show seems to be independent of religious identity. According to self-determination theory, humans have three basic psychological needs to be motivated towards an outcome including competence, belonging and autonomy [80]. While belonging and competence are likely to be affected by students' identities [81–83], our data support that any student regardless of their religious identity would benefit from perceiving that they have control over their own decisions to accept evolution. Since this is the first time that this construct has been shown to be relevant for improving evolution acceptance outcomes, there is much that remains to be explored about decision autonomy. For example, is the absence of forceful assertion that students need to accept evolution sufficient? Or do students need to be explicitly told that they have the decision to accept evolution? Further, does the extent to which the student trust the person telling them about evolution matter for their perception of autonomy?

These results on the beneficial impact of compatibility and autonomy instruction are one additional piece of evidence building on a long line of studies from multiple methodologies indicating that conflict reducing practices in evolution education can improve student evolution acceptance outcomes. Students report in interviews [15, 21] and when journaling [60] that when instructors implement practices considered to be conflict reducing that their evolution acceptance and engagement in learning evolution increases. These studies are corroborated by studies showing that student evolution acceptance increased when instructors implemented conflict reducing practices in a single course [17, 84] and when they implemented these practices in multiple courses [16, 18, 59, 85]. Now, our study shows that when we compare students that perceive different levels of conflict reducing practices, specifically compatibility and decision autonomy, we also show evidence for improved student evolution acceptance when they perceive higher levels of these practices.

## Future research

We argue that in light of the mounting evidence for conflict reducing practices that we as researchers need to move beyond studying *whether* conflict reducing practices work to what are the *most effective ways* to implement conflict reducing practices, which will warrant additional controlled study designs. Below we highlight how future research should move towards specifying (1) which conflict reducing practices are most effective, (2) what is the most effective implementation of these conflict reducing practices, and (3) whether practices are differentially effective based on student identity. Detailed qualitative, quantitative, and mixed methods research have successfully identified conflict reducing practices that students report improve their evolution acceptance including compatibility practices such as providing examples of religious individuals and biologist who accept evolution [15, 17, 18, 20], discussing the bounded nature of science [52–55, 58, 86–88] and discussing how evolutionary theory is agnostic about the existence or influence of a God/god(s) [7], as well as inclusive teaching practices such as remaining respectful of students' religious identities [19, 35, 89], giving students opportunities to reflect on their positions on evolution [19, 35, 49, 59, 90, 91] and now, fostering a sense of autonomy over their decision whether to accept evolution. In this study, we were able to measure student experiences of two of these practices, providing examples of compatibility through role models and autonomy, and show that when we compare students with different perceived levels of experience with these practices, that students who do perceive more agreement with the presence of the practices have better outcomes. However, we were unable to create a survey measure that had sufficient validity evidence that measured these other practices. A controlled study design would allow future researchers to implement these

practices in standardized and varied ways and determine which practices may be most impactful and what are the most effective ways to implement them for students with different religious and non-religious identities. With controlled studies in which students are randomly assigned to instruction that is identical except for one factor (e.g., the type of role model), we can more precisely start to characterize best and most effective ways to implement conflict reducing practices.

## Conclusions

Despite wide recommendations for the use of evolution education practices that are conflict reducing for religious students, our study is the first to show efficacy of these practices by comparing student experiences and outcomes across a multitude of classes with variability in both students and instructional practices. Using this large and geographically variable dataset, we found that for highly religious Christian students, when instructors provide examples of compatibility between religion and evolution, this led to higher evolution acceptance after instruction. Secondly, we showed that when students reported that their instructors gave them autonomy over their decisions to accept evolution, it led to greater gains in students' human evolution acceptance regardless of their religious affiliation or religiosity levels. We posit that the evidence for the efficacy of using conflict reducing practices to improve student outcomes has reached a critical threshold for which it will now be difficult for evolution instructors to justify excluding these practices from their evolution instruction if they wish to increase student evolution acceptance in introductory college biology classes. Future research should move beyond asking whether conflict reducing practices in evolution education can be effective to better understanding how they work and how to best implement these practices for students from a variety of religious and non-religious identities. Data indicates that if instructors were to implement conflict reducing practices more regularly in undergraduate biology education, evolution acceptance rates will rise faster.

## Supporting information

**S1 Appendix.**
(XLSX)

**S1 Text. Survey items used in analyses and final outcomes.**
(DOCX)

**S2 Text. Item fit statistics for partial credit Rasch models.**
(DOCX)

**S1 Table. Response rate by course.**
(DOCX)

**S2 Table. Details of Rasch transformation of Likert response options.**
(DOCX)

**S3 Table. Estimated marginal means with standard error from a linear mixed model.** Estimated marginal means with standard error from a linear mixed model for human evolution acceptance among undergraduate biology students with different religious affiliations. Results are averaged over the levels of gender, race, and biology major.
(DOCX)

**S1 Fig. Wright Map and Item Characteristic Curve of religious role model measures.** The histogram on the left represents the distribution of individuals' "ability", or test taker'

attributes. The score of individual ability was standardized in logits with 0 as the average and a higher value indicates a student agreed more compared to other students. The data point on the right represents item "difficulties", or items' attribute. The higher point indicates more "difficult" items or more disagreement. The higher point indicates more "difficult" items or more disagreement. For example, Cat6 is "strongly disagree" and Cat1 is "strongly agree". (TIF)

**S2 Fig. Wright Map and Item Characteristic Curve of autonomy measures.** The histogram on the left represents the distribution of individuals' "ability", or test taker attributes. The score of individual ability was standardized in logits with 0 as the average and a higher value indicates a student agreed more compared to other students. The data point on the right represents item "difficulties", or items' attribute. The higher point indicates more "difficult" items or more disagreement. For example, Cat6 is "strongly disagree" and Cat1 is "strongly agree". (TIF)

## Acknowledgments

We would like to acknowledge the instructors of biology courses willing to implement our survey and the thousands of students who were willing to complete the survey.

## Author Contributions

**Conceptualization:** Sara Brownell, M. Elizabeth Barnes.

**Data curation:** Hayley Dunlop, M. Elizabeth Barnes.

**Formal analysis:** Rahmi Qurota Aini, K. Supriya, Hayley Dunlop, Baylee Edwards, Samantha Maas, Julie Roberts, Alexa Summersill, Yi Zheng, M. Elizabeth Barnes.

**Funding acquisition:** Sara Brownell, M. Elizabeth Barnes.

**Methodology:** Rahmi Qurota Aini, K. Supriya, Yi Zheng.

**Supervision:** Sara Brownell, M. Elizabeth Barnes.

**Visualization:** Rahmi Qurota Aini.

**Writing – original draft:** Rahmi Qurota Aini.

**Writing – review & editing:** K. Supriya, Hayley Dunlop, Baylee Edwards, Samantha Maas, Julie Roberts, Alexa Summersill, Yi Zheng, Sara Brownell, M. Elizabeth Barnes.

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
