## [Decision Letter · Decision Letter 0]

22 Aug 2024

PONE-D-24-17358Student perceptions of conflict reducing practices in evolution education are associated with increases in their evolution acceptance in a large naturalistic studyPLOS ONE

Dear Dr. Barnes,

Thank you for submitting your manuscript to PLOS ONE. After careful consideration, we feel that it has merit but does not fully meet PLOS ONE’s publication criteria as it currently stands. Therefore, we invite you to submit a revised version of the manuscript that addresses the points raised during the review process.

Below, I have outlined the revisions needed:

**Table and Figures:**

Table 1: The heading "Acceptance of evolution" may be misleading, as the percentages might not reflect this concept accurately.Figures 3 and 5:Clarify the categories as "more than 1 SD above/below the mean" and use labels like "≤ -1 SD" or "≥ +1 SD" for clarity.Resolve the inconsistency regarding the description of "students affiliated with no religion" by rephrasing the caption.Define what "mean" represents in the context of these figures to avoid confusion.Explain what the shaded areas in the plots signify, possibly the 95% CI for the linear model.Figure 4: Correct the dashed horizontal line, which should represent the "overall mean within all samples," and fix a copy-paste error in the R code.

**Abstract:**

Add the background of the study at the beginning.Include a statement about the potential transformative impact of the research on evolution education.Specify the implications of the findings for evolution education practices.

**Introduction:**

Clearly state the research questions and objectives in a dedicated paragraph.Include a brief discussion of the theoretical framework, particularly cultural competence in science education.

**Methods:**

Improve the clarity of the statistical analysis plan by using subheadings and explaining the rationale for each control variable.Address how the sampling methods ensure the representativeness of the sample and discuss the adequacy of the 44% response rate.

**Results:**

Use subheadings to separate results for each research question or hypothesis.Include effect sizes along with p-values to provide a sense of the magnitude of observed effects.Briefly mention the statistical methods used beneath the figures and tables.

**Discussion:**

Add more depth to the discussion section, particularly in interpreting the results.Address limitations more explicitly, especially the sample size for non-Christian religious groups.Expand the discussion on why religious role models were effective only for Christian students, considering cultural and contextual factors.Discuss the potential long-term effects of the conflict-reducing practices and their relevance to the field.Consider adding a section on ethical considerations, including informed consent.

**Title and Tone:**

Shorten the title to make it more concise.Add an introductory sentence in the abstract to address the research problem.Minimize the use of first-person pronouns to enhance the academic tone.Clarify the findings presented in the abstract to reflect all data sets collected.

**General Suggestions:**

Ensure that the discussion includes more in-depth analysis of the findings rather than being dominated by comparisons with previous studies.Consider reorganizing the methodology section for better flow, particularly regarding the construct validation paragraph. ==============================

We look forward to receiving your revised manuscript.

Kind regards,

Mc Rollyn Daquiado Vallespin

Academic Editor

PLOS ONE

Journal Requirements: 

"NSF Grant number 1818659"

3. Please expand the acronym “NSF” (as indicated in your financial disclosure) so that it states the name of your funders in full.

"We would like to acknowledge the instructors of biology courses willing to implement our survey and the thousands of students who were willing to complete the survey. This project was supported by the NSF Grant number 1818659."

"NSF Grant number 1818659"

Reviewers' comments:

Reviewer's Responses to Questions

**Comments to the Author**

1. Is the manuscript technically sound, and do the data support the conclusions?

Reviewer #1: Yes

Reviewer #2: Yes

Reviewer #3: Yes

2. Has the statistical analysis been performed appropriately and rigorously? 

Reviewer #1: I Don't Know

Reviewer #2: I Don't Know

Reviewer #3: Yes

3. Have the authors made all data underlying the findings in their manuscript fully available?

Reviewer #1: Yes

Reviewer #2: Yes

Reviewer #3: Yes

4. Is the manuscript presented in an intelligible fashion and written in standard English?

Reviewer #1: Yes

Reviewer #2: Yes

Reviewer #3: Yes

5. Review Comments to the Author

Reviewer #1: This is a well-written, timely, and thoughtful research paper. The methodology, subject selection, and survey development all seem to be excellent.

There are a few small errors/inconsistencies that I happened to notice which should be corrected-- and I would ask that the authors please double-check all of the statistical analyses to ensure there aren't any additional errors. My concern is that a few sloppy (but minor) errors might indicate that something else has been overlooked.

Specific things that I happened to notice:

** In Table 1, there is a heading that says "Acceptance of evolution" which doesn't seem correct - I don't think those percentages indicate "acceptance of evolution."

** Figures 3 and 5 are not described completely or correctly. For instance:

- I think the categories should be described as "more that 1 SD above the mean" or "more than 1 SD below" instead of "1 SD above" or "1 SD below"? In the figure legend perhaps it would be clearer with a label like "≤ -1 SD" or "≥ +1 SD" ?

- The caption says "The red lines are students affiliated with no religion and have religiosity measures 1 SD below the mean." but in the text it states "students affiliated with no religion" in fact "may have checked the box for “Christian”, “Muslim”, “Jewish”, etc, we considered them non-religious because they did not believe this was a relevant or important part of their identities" So there seems to be some inconsistency there. Perhaps it would be better to state "The red lines are students with religiosity measures more than 1 SD below the mean, which we have coded as having no religious affiliation."

- Do the dashed lines that say "mean" represent the mean of the entire population of individuals coded in the specific subgroups (i.e. the mean of all students who identified as "Christian" or as "Other-Religion")? There are several uses of the term "mean" in the figure and caption and I can't always tell what is meant in each case.

- I couldn't find any description of what the shaded areas on these plots signifies— is it 95% CI for the linear model?

** In Figure 4 the dashed horizontal line should represent the "overall mean within all samples" but that obviously can't be correct. I found a simple copy-paste error in the R code (thank you SO MUCH for sharing your code! excellent.) - in line 379 in the file on GitHub it plots "course_stats_rsrm$mean" but should be plotting "course_stats_aut$mean"

I haven't tried to re-run the R code myself or repeat any of the analyses, but I would ask that the authors, in preparing a re-submission to correct these small points, please also ensure that there aren't other errors that I might not have noticed in my review of the manuscript. I think that the statistical analyses are appropriate and correct, but the small concerns identified above make me hesitant to give it my full confidence.

The discussion and survey of prior work in this area are thorough, and well-written. Excellent work!

Reviewer #2: Dear Editor,

Rahmi et al. have conducted a significant study on Student perceptions of conflict-reducing practices in evolution education. Their research, which is associated with increases in evolution acceptance, is a unique and valuable contribution to the field. With some major revisions, this study has the potential to become an even stronger piece of research.

Abstract:

The abstract provides a clear overview of the study's purpose, methods, and key findings. However, there are a few areas that could be improved.

1. Add the background of the study at the beginning of the abstract.

2. Consider adding a brief statement about the potential transformative impact of this research in the broader context of evolution education. This could inspire further research and innovation in the field.

3. The conclusion could be more specific about the implications of these findings for evolution education practices.

Introduction

1. The research questions are the backbone of the study and could be more clearly stated. Consider adding a paragraph that explicitly outlines the study's objectives and hypotheses, emphasizing their importance to the research process.

2. The introduction could benefit from a brief discussion of the theoretical framework guiding this research, particularly regarding the concept of cultural competence in science education.

Material methods

The statistical analysis plan is appropriate, but it could be more clearly presented. Consider using subheadings to separate the different analyses performed and provide a brief rationale for the inclusion of each control variable.

Results

The results are presented clearly and correspond well to the research questions. However, there are a few suggestions for improvement:

1. Consider using subheadings to separate the results for each research question or hypothesis.

2. It would be helpful to include effect sizes along with p-values to give readers a better sense of the magnitude of the observed effects.

3. Briefly mention the statistical methods used to derive key figures and results beneath the figures and tables.

Discussion

1. The discussion section provides a general interpretation of the results. While it lacks depth in explanation, it is evident that a significant amount of effort has been put into this section.

2. The analysis of differential effects across religious affiliations is somewhat limited by sample size for non-Christian religious groups. This limitation, along with others, could be addressed more explicitly in the limitations section, providing a more comprehensive understanding of the research's scope and potential areas for future investigation.

3. The discussion of why religious role models were only effective for Christian students could be expanded, perhaps considering cultural and contextual factors more deeply.

4. The potential long-term effects of these conflict-reducing practices are not addressed. A more detailed discussion of whether these changes in evolution acceptance persist over time would be valuable, as it would highlight the potential implications of the research and its relevance to the field.

5. Consider adding a brief section on ethical considerations, including how informed consent was obtained from participants.

Regards,

Reviewer #3: Title:

Your title is too long. It would be better if it could be shortened.

Abstract:

Please add an introductory sentence at the beginning of the abstract that addresses the research problem you are investigating.

You have used first-person pronouns excessively. To enhance the academic tone of the manuscript, minimize the use of first-person pronouns.

Based on the second and third sentences, it appears that you collected at least three sets of data. However, the fourth sentence only presents one finding.

Introduction:

The introduction is well-structured with a clear flow, allowing readers to follow the developed framework and the issues being reported effectively.

Although brief, the state-of-the-art and research gap related to the study’s topic are addressed in the fourth paragraph.

Methods:

The research methods are presented comprehensively, with sufficient detail and clarity, allowing readers to fully understand the design and steps of the study, from instrument development to the analysis of findings.

One minor suggestion for making the methodology section more systematic is to place the paragraph on construct validation for the survey instrument measuring Conflict Reducing Practices in Evolution Education (predictor variable) after the paragraph on expert and student reviews and before the paragraph on class observation.

You used convenience and snowball sampling methods. How did you ensure that the sample is representative of the population?

Is a 44% response rate considered adequate?

Results:

The research findings are presented in a complete and systematic manner.

Discussion:

While the discussion addresses the findings, it lacks depth and breadth. The discussion is currently dominated by comparisons between your findings and those of previous studies, as well as suggestions for future research based on unanswered questions. These two components should not be reduced in the revised manuscript, but you should add more in-depth discussion of the findings you have obtained.

6. PLOS authors have the option to publish the peer review history of their article (what does this mean?). If published, this will include your full peer review and any attached files.

Reviewer #1: No

Reviewer #2: No

Reviewer #3: **Yes: **Ahmad Fauzi

---

## [Author Response · Author response to Decision Letter 0]

4 Oct 2024

Dear monitoring editor and reviewers,

Thank you for your detailed and thoughtful review of our manuscript, "Conflict-reducing practices in evolution education are associated with increases in evolution acceptance: A large naturalistic study," for further consideration in PLOS ONE. We have carefully considered the feedback and believe we have significantly improved the manuscript.

First, we expanded the discussion to enhance the manuscript's potential for impact. We also expanded the description of the theoretical framework, particularly adding clarity to cultural competence as a frame. Additionally, we have made the limitations of our research more explicit, particularly regarding our inability to address the distinct experiences and perspectives of individual religious groups that are not Christian. We have also addressed the potential limitation of the generalizability of our sample. Another key revision involved improving the accuracy and clarity of our results, which reflected in the figures, figure captions, tables, and subheadings. 

We have responded to each reviewer’s comments, as outlined by the editor, in more detail in "Response to reviewers.doc". We hope these revisions bring the work closer to being suitable for publication.

---

## [Editor Report · Decision Letter 1]

25 Oct 2024

Conflict reducing practices in evolution education are associated with increases in evolution acceptance in a large naturalistic study

PONE-D-24-17358R1

Dear Dr. BARNES,

We’re pleased to inform you that your manuscript has been judged scientifically suitable for publication and will be formally accepted for publication once it meets all outstanding technical requirements.

Kind regards,

Mc Rollyn Daquiado Vallespin

Academic Editor

PLOS ONE
---

## [Editor Report · Acceptance letter]

4 Nov 2024

PONE-D-24-17358R1 

PLOS ONE

Dear Dr. Barnes, 

I'm pleased to inform you that your manuscript has been deemed suitable for publication in PLOS ONE. Congratulations! Your manuscript is now being handed over to our production team.

Kind regards, 

on behalf of

Dr. Mc Rollyn Daquiado Vallespin 

Academic Editor

PLOS ONE